# Asymmetric $\alpha$-allylic allenylation of $\beta$-ketocarbonyls and aldehydes by synergistic Pd/chiral primary amine catalysis

Chang You[1], Mingying Shi[2], Xueling Mi[2] ✉ & Sanzhong Luo ✉[1] ✉

We herein describe an asymmetric $\alpha$-allylic allenylation of $\beta$-ketocarbonyls and aldehydes with 1,3-enynes. A synergistic chiral primary amine/Pd catalyst was identified to facilitate the utilization of 1,3-enynes as atom-economic and achiral allene precursors. The synergistic catalysis enables the construction of all-carbon quaternary centers-tethered allenes bearing non-adjacent 1,3-axial central stereogenic centers in high level of diastereo- and enantio-selectivity. By switching the configurations of ligands and aminocatalysts, diastereodivergence can be achieved and any of the four diastereoisomers can be accessed in high diastereo- and enantio- selectivity.

Enantioselective $\alpha$-allylic allenylation of carbonyls is a powerful and straightforward approach to access chiral allenes that are of significant relevance as synthons and bio-active products[1–8]. Comparing with the typical allylic alkylation counterparts, allenylation process poses an additional stereochemical issue in controlling the axial chirality of allene beyond the stereogenic center construction[9–21]. A more challenging stereochemical situation is the concurrent formation of non-adjacent 1,3-axial central stereogenic centers in the $\alpha$-allenylation of carbonyls (Fig. 1I)[22–24]. Progresses along this line are rather limited. Recently, enantioselective $\alpha$-allylic allenylation of $\alpha$-hydoxyl ketones and imine esters have been achieved with racemic allenylic esters by the work of Trost, Ma and Zhang, respectively[22,24]. The group of He used 1,3-enynes as atom-economic and achiral allene precursors for asymmetric allylic allenylation of $\alpha$-fluoroketones[23]. Despite of these advances, the reactions are limited to those functionalized esters or ketones bearing additional coordinating groups. The reactions with $\beta$-ketocarbonyls and aldehydes remains underdeveloped and there is no example in the formation of all-carbon quaternary centers for this type of allylic allenylation process.

Previously, we have shown metal-hydride (M-H) cycle could work in concert with chiral primary aminocatalysis to facilitate enantioselective alkylation with allene, 1,3-dienes and alkynes[25–30]. In this work, we report synergistic Pd/chiral primary amine catalysis for $\alpha$-allylic allenylation of $\beta$-ketocarbonyls and aldehydes with the most atom-economic and achiral allene precursors, 1,3-enynes (Fig. 1II). Distinctive from the dual metal synergistic catalysis, the current organo-metal dual catalytic system eliminates the necessity on coordinating groups, and thus significantly expands the scopes to include versatile $\beta$-ketocarbonyls and aldehydes for the first time. The reaction also features the construction of all-carbon quaternary centers-tethered allenes bearing a non-adjacent 1,3-axial-central chirality, a challenging target that has not been achieved before.

## Results and discussion
### Reaction optimizations
Based on our preliminary successes on achiral Pd/chiral primary amine catalysis[25–30], conjugated enyne **1a** and racemic *tert*-butyl 2-methyl-3-oxobutanoate **2a** were selected as the model substrates to examine the feasibility of this design in the presence of chiral primary-tertiary (S)-**A1** amine catalyst, Pd$_2$(dba)$_3$ and DPEphos in the DCE at 40 °C (Table 1). A set of chiral primary-tertiary diamine catalysts were first examined, and **A1** was found to provide **3a** with 90% *ee*, although the yield was only 51% (entry 2). Other bulkier catalysts such as **A2** slightly increased the enantioselectivity but with compromised activity (entry 3 vs entry 2). A survey of classic biphosphine ligands revealed that both of the reactivity and stereoselectivity were significantly influenced by the dihedral angle of the ligand backbone. Chiral ligand **L4** with the smallest dihedral angle was identified as an efficient ligand in terms of both reactivity and stereoselectivity (entries 4–6), whereas the more sterically hindered ligand **L5** gave the desired adduct with >99% *ee*, albeit with only trace product. Ligands **L6** and **L7**, which were effective in other metal-hydride (M-H) mediated reactions[31–33], turned out to be

[1]Center of Basic Molecular Science, Department of Chemistry, Tsinghua University, Beijing 100084, China. [2]College of Chemistry, Beijing Normal University, Beijing 100875, China. ✉e-mail: xlmi@bnu.edu.cn; luosz@tsinghua.edu.cn

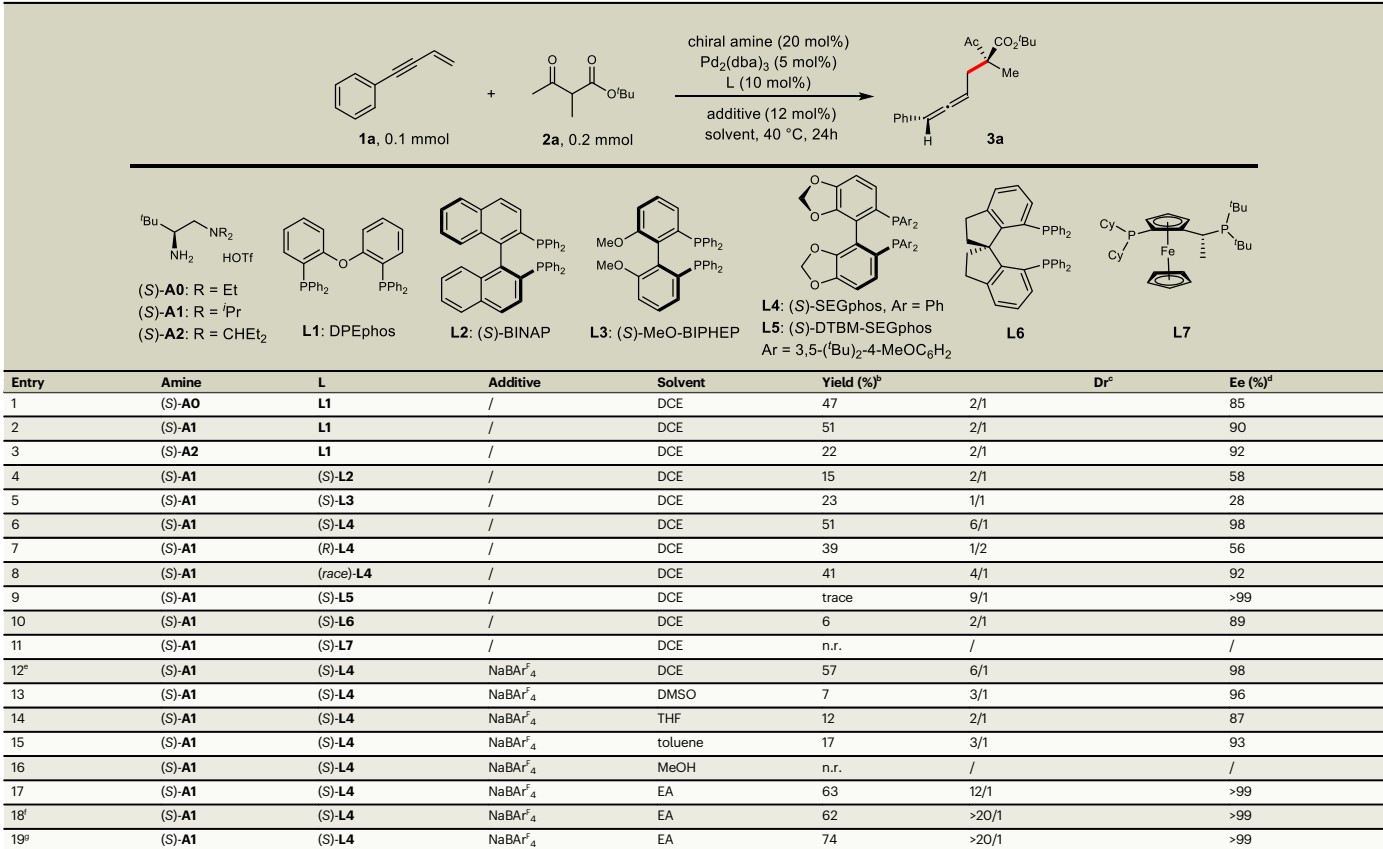

**Fig. 1 | Catalytic asymmetric α-allylic allenylation. I** Previous strategies on asymmetric α-Allylic allenylation of carbonyls. **II** Synergist amine/Pd catalysis for allenylation of ketones and aldehydes presented in this work.

## Table 1 | Screening and optimization[a]

| Entry | Amine | L | Additive | Solvent | Yield (%)[b] | Dr[c] | Ee (%)[d] |
|---|---|---|---|---|---|---|---|
| 1 | (S)-**A0** | **L1** | / | DCE | 47 | 2/1 | 85 |
| 2 | (S)-**A1** | **L1** | / | DCE | 51 | 2/1 | 90 |
| 3 | (S)-**A2** | **L1** | / | DCE | 22 | 2/1 | 92 |
| 4 | (S)-**A1** | (S)-**L2** | / | DCE | 15 | 2/1 | 58 |
| 5 | (S)-**A1** | (S)-**L3** | / | DCE | 23 | 1/1 | 28 |
| 6 | (S)-**A1** | (S)-**L4** | / | DCE | 51 | 6/1 | 98 |
| 7 | (S)-**A1** | (R)-**L4** | / | DCE | 39 | 1/2 | 56 |
| 8 | (S)-**A1** | (race)-**L4** | / | DCE | 41 | 4/1 | 92 |
| 9 | (S)-**A1** | (S)-**L5** | / | DCE | trace | 9/1 | >99 |
| 10 | (S)-**A1** | (S)-**L6** | / | DCE | 6 | 2/1 | 89 |
| 11 | (S)-**A1** | (S)-**L7** | / | DCE | n.r. | / | / |
| 12[e] | (S)-**A1** | (S)-**L4** | NaBAr$^F_4$ | DCE | 57 | 6/1 | 98 |
| 13 | (S)-**A1** | (S)-**L4** | NaBAr$^F_4$ | DMSO | 7 | 3/1 | 96 |
| 14 | (S)-**A1** | (S)-**L4** | NaBAr$^F_4$ | THF | 12 | 2/1 | 87 |
| 15 | (S)-**A1** | (S)-**L4** | NaBAr$^F_4$ | toluene | 17 | 3/1 | 93 |
| 16 | (S)-**A1** | (S)-**L4** | NaBAr$^F_4$ | MeOH | n.r. | / | / |
| 17 | (S)-**A1** | (S)-**L4** | NaBAr$^F_4$ | EA | 63 | 12/1 | >99 |
| 18[f] | (S)-**A1** | (S)-**L4** | NaBAr$^F_4$ | EA | 62 | >20/1 | >99 |
| 19[g] | (S)-**A1** | (S)-**L4** | NaBAr$^F_4$ | EA | 74 | >20/1 | >99 |

[a]Reaction conditions, unless otherwise noted: **1a** (0.1 mmol), **2a** (0.2 mmol), Pd₂(dba)₃ (5 mol %), Ligand (10 mol %), amine* (20 mol%), and additive (12 mol%) in 1,2-dichloroethane (DCE, 0.3 mL) at 40 °C for 24 h.
[b]Isolated yield of all diastereoisomers; n.r. = no reaction.
[c]Dr values were determined by ¹HNMR analysis.
[d]The ee values were determined by chiral HPLC analysis of the alcohol derivative of the product.
[e]NaBAr$^F_4$ (Ar$^F$ = 3, 5-(CF₃)₂Ph) as a promoting additive.
[f]Pd(PPh₃)₄ instead of Pd₂(dba)₃.
[g]NHTf₂ instead of TfOH.

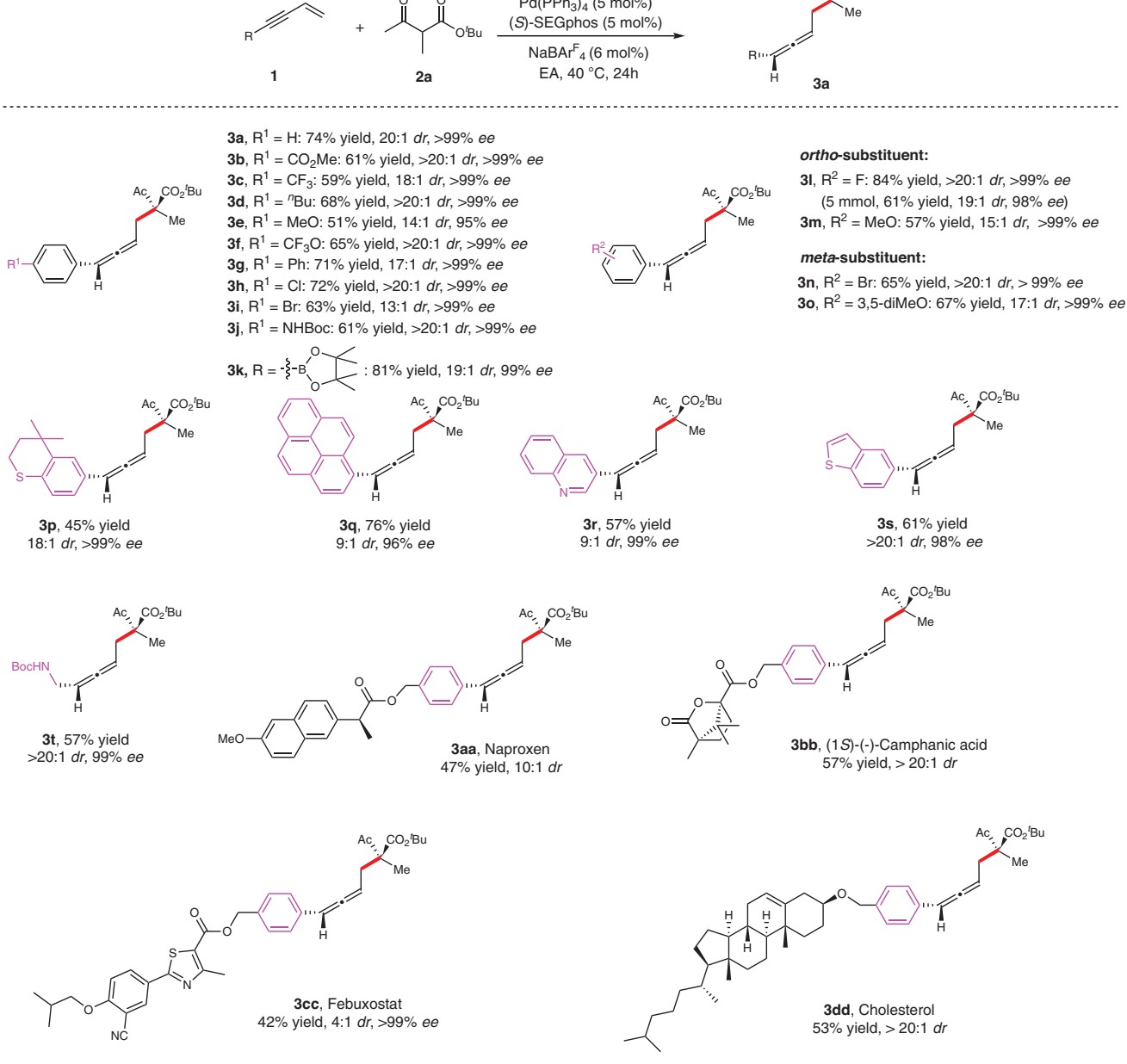

**Fig. 2 | Scopes of 1,3-enynes.** Reaction conditions, unless otherwise noted: **1** (0.1 mmol), **2a** (0.3 mmol), Pd(PPh₃)₄ (5 mol %), **L4** (5 mol %), **A1** (20 mol %), and NaBArᶠ₄ (6 mol %) in EA (0.3 mL) at 40 °C for 24 h.

inactive in our Pd/aminocatalytic system (entries 10 and 11). Match-mismatch effect was noted on the ligand configuration. The use of (*R*)-**L4**/(*S*)-**A1** led to a reversed diastereoselectivity, with unfortunately reduced activity and enantioselectivity (entry 6 vs entry 7). In addition, (*race*)-**L4**/(*S*)-**A1** showed only slightly diminished stereoselectivity, suggesting that the chirality of ligand has only marginal effect on enantioselectivity and that aminocatalyst plays a major role in stereocontrol (Table 1, entry 8 vs entry 6). Further optimization identified NaBArᶠ₄ (Arᶠ = 3, 5-(CF₃)₂Ph) as a promoting additive (Table 1, entry 12) and ethyl acetate (EA) as the selection of solvent for high productivity (Table 1, entries 13–17). When changing Pd₂(dba)₃ to Pd(PPh₃)₄, the desired product was obtained with high diastereoselectivity (Table 1, entry 18). Under these conditions, the swap of strong acid additive TfOH with NHTf₂ further increased the yield to 74%, meanwhile maintaining >20:1 *dr* and >99% *ee* (Table 1, entry 19).

With the optimized conditions in hand, we explored the scopes of enynes by carrying out reactions with **2a**, and the results are summarized in Fig. 2. A diverse array of electron-withdrawing and electron-donating groups at varying positions on the aryl units were found to be well tolerated to afford the allene products (Fig. 2, **3a–o**) in 51–84% yield with excellent stereoselectivities (13:1 to >20:1 *dr* and exclusively >99% *ee*). Particularly, the enyne **1i** with bromo-substitution gave the expected product **3i** in 63% yield with 13:1 *dr* and >99% *ee* without obvious side products associated with the bromo-substitution in the presence of Pd. Multi-substituted arenes were also suitable as well to deliver the corresponding products in good yields with good diastereoselectivities and excellent enantioselectivities (Fig. 2, **3o, p**). Heteroaryl enynes including those highly conjugate arenes were compatible with this process, providing the allene products (Fig. 2, **3q–s**) in 57-76% yield with 9:1 to >20:1 *dr* and up to >99% *ee*. In addition,

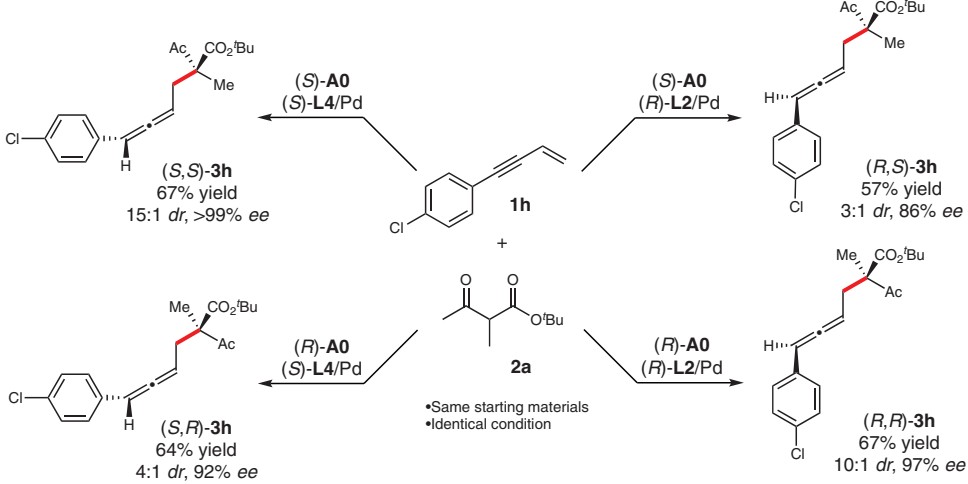

**Fig. 3 | Scopes of β-ketocarbonyls and aldehydes.** Reaction conditions, unless otherwise noted: **1 l** (0.1 mmol), **2** (0.3 mmol), Pd(PPh₃)₄ (5 mol %), **L4** (5 mol %), **A1** (20 mol %), and NaBAr^F_4 (6 mol %) in EA (0.3 mL) at 40 °C for 24 h.

**Fig. 4 | Stereodivergent synthesis.** Stereodivergent synthesis of all the four stereoisomers of **3 h** by the combination of **A0** and **L4/L2** with varied configurations.

alkyl-substituted enyne also showed moderate coupling reactivity with >20:1 *dr* and >99% *ee* (Fig. 2, **3t**).

The practicality of this method was examined in the late-stage allenylation of structurally complexed substrates bearing diverse biologically active molecules. For example, the reactions

worked smoothly with enynes tethered to drugs such as Naproxen (**3aa**), (1S)-(-)-Camphanic acid (**3bb**), Febuxostat (**3cc**) and Cholesterol (**3dd**), to give the desired adducts with moderate reactivity and high diastereo- and enantioselectivities (Fig. 2).

We also examined the scope of β-ketocarbonyls. As show in Fig. 3A, different ester groups with varied sizes were generally tolerated to give the desired products in good yields and excellent stereoselectivities (Fig. 3A, **4a–d**). Large α-substituent led to reduced reactivity and stereoselectivity (Fig. 3A, **4f**), while small one worked well to give the desired adduct (Fig. 3A, **4e**) in 11:1 *dr*, >99% *ee* and 64% yield. The reaction with cyclic β-ketoesters such as cyclopentanone and cyclohexanone worked well to deliver the corresponding products with excellent stereoselectivities but in lower yields (Fig. 3A and 4g–i), most likely due to the lower reactivity of these ketones.

Next, the scope of β-ketoamides was evaluated for this process (Fig. 3B). For acylic β-ketoamides, the yields were decreased slightly but with significantly higher *dr* comparing with cyclic β-ketoamide (Fig. 3B, **4k–l vs 4m–p**). To our delight, cyclic β-ketoamides turned to be favorable substrates for the reactions, showing higher activity than their β-ketoester counterparts (Fig. 3B, **4n–p vs 4h–i**). Different amide groups were tolerated to give good yields and high stereoselectivities (Fig. 3B, **4j, 4n–p**).

We then explored the reactions with α-branched aldehydes (Fig. 3C), for which there has been no examples on asymmetric allylic allenylation. α-Arylpropanals can be well applied and the reactions proceeded smoothly to give the desired adducts with good yields and high stereoselectivity (Fig. 3C, **4q–t**). The diastereoselectivity was relatively lower comparing with that of the reactions with β-ketocarbonyls.

## Synthetic applications

Stereodivergent catalysis has recently become a powerful strategy to access any stereoisomers of chiral molecules bearing multi-stereogenic centers in a highly controlled manner[34–47]. Early successes along this line have been achieved with chiral aminocatalysts[35]. In our cases, the initially identified bulky aminocatalyst **A1** showed no divergence on diastereoselectivity and the switch to other configured chiral ligand resulted in only poor diastereoselectivity (Table 1 entry 7). After extensive investigation, we found the less bulky catalyst **A0** showed promising results (Fig. 4) and the combination of **A0** and **L4/L2** with varied configurations led to any of the four diastereoisomers in good yields, high enantioselectivities and with moderate to good diastereoselectivities (up to 67% yield, 15:1 *dr*, and >99% *ee*).

The reaction between **1l** and **2a** could be conducted at a gram scale with comparable results (Fig. 2, **3l**), demonstrating the practicality of the current reaction. In addition, this series of products are rich in functionality for further transformations that were enormously demonstrated[48–52]. In our cases, the initially obtained allylic allene adducts could be easily transformed in order to determine the stereochemical configurations. Upon treatment of **3a** with 2,4-dinitrophenylhydrazine followed by HCl (1 M, 1.0 equiv.) in MeOH led to compound **5a** in 67% yield and 19:1 *dr* with 99% *ee* (Fig. 5, **5a**). Reduction of the keto moiety of (*S,S*)-**3a** over NaBH₄ gave **5b** in 98% yield (Fig. 5, **5b**). The absolute configuration of reduction product **5b** was assigned based on crystallographic analysis.

## Proposed catalytic cycle

In accordance with previous studies[23,25–30], a synergistic Pd/aminocatalytic cycle could be proposed as described in Fig. 6. For palladium catalysis, a L*/Pd-H complex generated in situ underwent hydrocarbonation with enyne regioselectively to give the intermediate **I**, which could interconvert rapidly to the key π-allyl-palladium syn-η³-int **II** via σ-π rearrangement[15,53,54]. The protonated aminocatalyst may serve as the proton source for Pd-H formation. Subsequently, intermediate syn-η³-Int **II** coupled with the enamine intermediate **III** to give the desired product **3a** after hydrolysis. On the basis of experimental observations as well as our previous studies[27], a steric model could be proposed to account for the observed stereoselectivity regarding the chiral center construction (Fig. 6, **III**), whereas chiral ligand in synergy with aminocatalysts plays the key role in steering the axial selectivity.

In summary, we have developed a synergistic chiral primary amine/Pd catalysis for α-allylic allenylation of ketones and aldehydes. The catalysis enables the use of 1,3-enynes as the atom-economic and achiral allene precursors, and features high

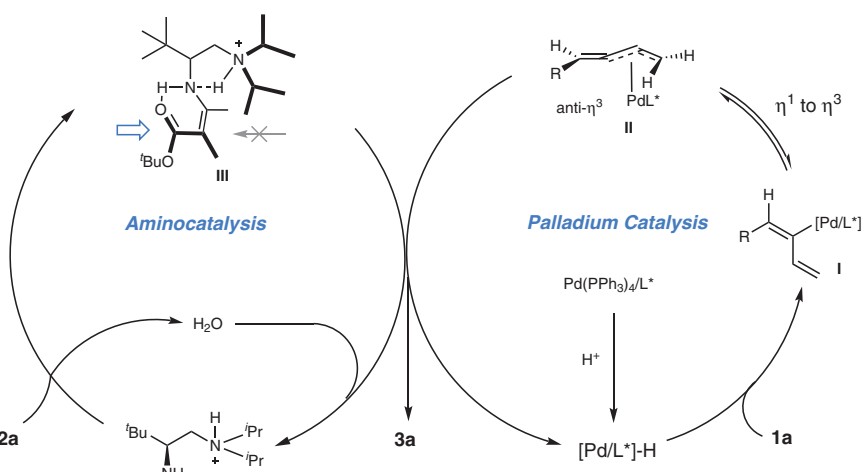

**Fig. 5 | Synthetic transformations and X-ray crystallography data of products.** Reaction conditions: **a** 2,4-dinitrophenylhydrazine (1.2 equiv.), 1 M HCl (1.0 equiv.), MeOH (0.1 M), r.t. **b** NaBH₄, MeOH (0.5 M), r.t., 2 h.

**Fig. 6 | Proposed mechanism.** Synergistic catalytic cycle of aminocatalysis and palladium catalysis.

stereoselectivity in the construction of non-adjacent 1,3-axial central chirality bearing all-carbon quaternary centers. This method could be generally applied to *β*-ketocarbonyls and aldehydes to afford the allenylation products with high enantioselectivity and diastereoselectivity. Divergence on the stereoselectivity could be achieved by varying the configurations of chiral primary amine and chiral ligands.

## Methods
### General procedure
In glove box, to a flame-dried Schlenk tube equipped with a magnetic stir bar was added *tert*-butyl 2-methyl-3-oxobutanoate (**2a**, 0.30 mmol), but-3-en-1-yn-1-ylbenzene (**1a**, 0.10 mmol), Pd(PPh$_3$)$_4$ (5 mol%), (*S*)-SEGphos (5 mol%), NaBAr$^F_4$ (6 mol%) and primary amine (*S*)-**A1** (20 mol %), the mixture was diluted with 0.3 mL of anhydrous EA, then the mixture was moved out from glove box and stirred at 40 °C for 24 h, solvent was removed and residue was purified by silica gel chromatography (10% EtOAc in Petroleum ether) to give **3a** as a colorless oil. The enantiometric excess was determined by HPLC (AD-H*2).

## Data availability
All data are available from the authors upon request. Supplementary information and chemical compound information are available along with the online version of the paper. The X-ray crystallographic coordinates for structures reported in this study have been deposited at the Cambridge Crystallographic Data Centre (CCDC) under deposition number 2222345 (**5b**). These data can be obtained free of charge from The Cambridge Crystallographic Data Centre via www.ccdc.cam.ac.uk/structures.

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

## Acknowledgements

We thank the Natural Science Foundation of China (21861132003 and 22031006 to S.L.) and Tsinghua University Initiative Scientific Research Program (to S.L.) for financial support.

## Author contributions

S.L. and X.M. conceived and directed the project. C.Y. optimized the reaction conditions, examined the substrate scope and studied the mechanism with the help of S.M. C.Y. and S. L. wrote the manuscript with contributions from all authors.

## Competing interests

The authors declare no competing interests.
