## [Peer Review File · Nature Communications]

REVIEWER COMMENTS

Reviewer #1 (Remarks to the Author):

This paper by Luo et al reports asymmetric hydroalkylation of conjugated enynes via synergistic palladium/organo catalysis. Moderate to good yields, high diastereo- and enantioselectivities are observed for a series of allene compounds bearing both a quaternary stereogenic center and a chiral axial moiety. Especially, they demonstrate the feasibility of stereodivergent preparations of all four stereoisomers of this type of compound, which is known to be very challenging and valuable to construct. This transformation provides a new route for the stereodivergent synthesis of skeletons containing two different types of chirality, especially show the potential of the combination of primary amine catalyst and metal hydride catalyst. Thus, I support the acceptance of this work after a major revision to address a series of questions and errors.

The method shows a reasonable substrate scope. I wonder the reason that many cases only provide moderate reaction yields. Is it caused by low reactivity of substrates or obvious side reactions? Related description might be added to the main text to provide guidance to the readers.

For the catalytic cycle, it seems that the authors think PdH was generated from the reaction of Pd(0) and H₂O. This might not be suitable, because HOTf or other proton sources might oxidize the low-valent palladium as well. The authors had better revise the figure to avoid the confusion. Or the authors need provide strong proof to support this proposal.

Some typos and errors need careful checking:

1) page 3, for sentence "conjugated enyne 1a and racemic tert-butyl 2-methyl-3-oxobutanoate 2a was selected as the model substrates", "1a" should be in bold; "tert" should be italic; "was" should be "were".

2) page 4, top sentence, "primary tertiary" is better to be written as "primary-tertiary".

3) what does "math-mismatch effect" mean? Is "match-mismatch"?

4) The results of prepared compounds 3aa, 3bb and 3dd in figure 2 need double-check. For compound 3aa, the corresponding enantioselectivity is unnecessary, due to the use of enantiopure substrate. For the latter two compounds, the ee values are missing.

5) "To our delights" is "To our delight".

6) "Different amides groups" should be "Different amide groups" or "Different amides".

7) "Earlies" should be "early" or use "prior".

8) "catalytic HCl (1 M, 1.0 equiv.)" could not be called catalytic.

9) In figure 2, "R" is used in the reaction scheme and also as a substituent of aryl unit. This might raise confusion due to different meanings of "R".

For the supporting information:

1) CDCl₃ at 77.16 ppm was chosen as the internal standard for carbon NMR described in the general information. However, all the labelled internal peaks were not 77.16. Thus, all the data and spectra need to be revised.

2) S88, S95 & S123, the HPLC figures of 3h & 3o need re-checking because of the inaccurate detection.

3) S100, the integration in the rac-HPLC figure of compound 3t is inaccurate, especially for the first two peaks of which the baseline is artificially tilted. As the integration values of these four peaks are very close to each other, this situation might lead to wrong judgement for the integration of chiral compound 3t.

4) S107. I do not know how the authors can determine the right peak positions for chiral compound, considering the difference of corresponding retention times between rac- and chiral compounds is heavily large. Although many others in this paper display similar situation, the judgement for retention time is basically reasonable and understandable.

5) Compound 5b is prepared in 19:1 dr. However, the corresponding carbon nmr does not meet it. Another set of carbon signals are not labelled but quite obvious.

Reviewer #2 (Remarks to the Author):

In this manuscript, Luo and Mi and their co-workers disclosed a synergistic Pd/amine system for constructing non-adjacent 1,3-axial-central chirality through hydroalkylation of 1,3-enyne with β -ketoester. Excellent enantio- and diastereoselectivity were obtained by using a combination of a chiral ligand with a chiral primary amine catalyst. For the kinetically unfavorable diastereoisomer, this method could also provide a moderate dr with excellent ee values by switching the configurations of ligand and aminocatalysts. Therefore, diastereodivergence can be achieved, although with low efficiency. Another impressive result is this system could be extended to α -

branched aldehydes with moderate diastereoselectivity but good enantioselectivity. Overall this is a nice piece of work that makes a good complement to the current systems to prepare non-adjacent 1,3-axial central chirality. I strongly recommend its acceptance in Nat. Commun. after some minor revisions.

1. Strictly speaking, β -ketoesters or β -ketoamides is not simple ketones. So please consider revising the title. Using β -ketocarboxyls instead of ketones might be OK, right?
2. In Fig. 1, the word "1,3-axial central chirality" is overlapped with other chemical structures. Please revise it.
3. Both the title and Fig.3C have indicated that aldehydes are part of the substrate scope, but I did not find any related description in the context. Am I missing something?
4. The π -allyl-palladium intermediate II is syn- η^3 rather than anti- η^3 . Please revise them accordingly throughout the manuscript.
5. Reference 22 is not completed.
6. Given that synergistic catalysis enabled stereodivergent synthesis is developed rapidly, some updated literature could be considered to cite.
 - 1) J. Am. Chem. Soc. 2021, 143, 10948-10962. In this paper, the amine/Pd catalyzed hydroalkylation of 1,3-dienes is achieved in a stereodivergent fashion.
 - 2) Nat. Commun. 2022, 13, 5876. In this paper, the first stereodivergent allylation of β -ketoesters was achieved by Ru/Pd synergistic catalysis.
 - 3) For organo/transition metal combined catalysis, an updated review paper, see: J. Am. Chem. Soc. 2022, 144, 2415–2437.

Some latest organo/transition metal catalyzed stereodivergent allylation:

Pd+PTC: Angew. Chem. Int. Ed. 2023, 62, e202215714

Pd+BTM: Chem 2022, 8, 2784-2796. and Angew. Chem. Int. Ed. 2022, 61, e202207621.

Responses and revisions

Reviewer #1 (Remarks to the Author):

This paper by Luo et al reports asymmetric hydroalkylation of conjugated enynes via synergistic palladium/organo catalysis. Moderate to good yields, high diastereo- and enantioselectivities are observed for a series of allene compounds bearing both a quaternary stereogenic center and a chiral axial moiety. Especially, they demonstrate the feasibility of stereodivergent preparations of all four stereoisomers of this type of compound, which is known to be very challenging and valuable to construct. This transformation provides a new route for the stereodivergent synthesis of skeletons containing two different types of chirality, especially show the potential of the combination of primary amine catalyst and metal hydride catalyst. Thus, I support the acceptance of this work after a major revision to address a series of questions and errors.

Many thanks for your comments.

Reviewer #1 (Remarks to the Author):

1. The method shows a reasonable substrate scope. I wonder the reason that many cases only provide moderate reaction yields. Is it caused by low reactivity of substrates or obvious side reactions? Related description might be added to the main text to provide guidance to the readers.

Reply: The low yields are generally due to the low reactivity of the ketones, particularly the cyclic ketocarboxyls, in this reaction as no obvious side products were detected. A brief discussion was added in the main text, p8.

2. For the catalytic cycle, it seems that the authors think PdH was generated from the reaction of Pd(0) and H₂O. This might not be suitable, because HOTf or other proton sources might oxidize the low-valent palladium as well. The authors had better revise the figure to avoid the confusion. Or the authors need provide strong proof to support this proposal.

Reply: A good point! We agree with the referee that the free acid (HNTf₂) in its conjugation with amine catalyst may serve as the proton source in the start-up stage. Water is required to fulfill aminocatalytic cycle. We modified the scheme to clarify this issue and a brief discussion is also added in the mechanism part.

Some typos and errors careful checking:

1. Page 3, for sentence “conjugated enyne 1a and racemic tert-butyl 2-methyl-3-oxobutanoate 2a was selected as the model substrates”, “1a” should be in bold; “tert” should be italic; “was” should be “were”.

Reply: These format issues have been corrected.

2. Page 4, top sentence, “primary tertiary” is better to be written as “primary-tertiary”.

Reply: Thanks, corrected.

3. What does “math-mismatch effect” mean? Is “match-mismatch”?

Reply: This spelling issue has been corrected.

4. The results of prepared compounds **3aa**, **3bb** and **3dd** in figure 2 need double-check. For compound **3aa**, the corresponding enantioselectivity is unnecessary, due to the use of enantiopure substrate. For the latter two compounds, the ee values are missing.

Reply: Thanks! Sorry about our oversight. These have been corrected. The ee value of **3aa** in the manuscript and SI were deleted. The ee value of **3dd** have been supplemented in the manuscript and the corresponding spectral data have been corrected in SI. We were unable to determine the ee of **3bb**. This compound was deleted and the numbering of compounds were re-ordered.

5. “To our delights” is “To our delight”

6. “Different amides groups” should be “Different amide groups” or “Different amides” .

7. “Earlies” should be “early” or use “prior” .

8. “catalytic HCl (1 M, 1.0 equiv.)” could not be called catalytic.

9. In figure 2, “R” is used in the reaction scheme and also as a substituent of aryl unit. This might raise confusion due to different meanings of “R” .

Reply: Many thanks. These have been corrected.

For the supporting information:

1. CDCl₃ at 77.16 ppm was chosen as the internal standard for carbon NMR described in the general information. However, all the labelled internal peaks were not 77.16. Thus, all the data and spectra need to be revised.

Reply: All internal standards have been changed to 77.16 and the corresponding spectral data has been corrected in SI.

2. S88, S95 & S123, the HPLC figures of **3h** & **3o** need re-checking because of the inaccurate detection.

Reply: The HPLC figures of **3h** & **3o** have been updated.

3. S100, the integration in the *rac*-HPLC figure of compound **3t** is inaccurate, especially for the first two peaks of which the baseline is artificially tilted. As the integration values of these four peaks are very close to each other, this situation might lead to wrong judgement for the integration of chiral compound **3t**.

Reply: We re-ran the HPLC of compound **3t**, and updated the data.

4. S107. I do not know how the authors can determine the right peak positions for chiral compound, considering the difference of corresponding retention times between *rac*- and chiral compounds is heavily large. Although many others in this paper display similar situation, the judgement for retention time is basically reasonable and understandable.

Reply: We changed HPLC separation method of **4f**, and the data were supplemented in the manuscript and SI.

4. Compound **5b** is prepared in 19:1 dr. However, the corresponding carbon nmr does not meet it. Another set of carbon signals are not labelled but quite obvious.

Reply: Although the chiral substrate **3a** was used, the reduction of keto moiety is not stereoselective, showing only 1:1 dr, which has been indicated in the manuscript and corresponding nmr in SI.

Reviewer #2 (Remarks to the Author):

In this manuscript, Luo and Mi and their co-workers disclosed a synergistic Pd/amine system for constructing non-adjacent 1,3-axial-central chirality through hydroalkylation of 1,3-enyne with β -ketoester. Excellent enantio- and diastereoselectivity were obtained by using a combination of a chiral ligand with a chiral primary amine catalyst. For the kinetically unfavorable diastereoisomer, this method could also provide a moderate dr with excellent ee values by switching the configurations of ligand and aminocatalysts. Therefore, diastereodivergence can be achieved, although with low efficiency. Another impressive result is this system could be extended to α -branched aldehydes with moderate diastereoselectivity but good enantioselectivity. Overall this is a nice piece of work that makes a good complement to the current systems to prepare non-adjacent 1,3-axial central chirality. I strongly recommend its acceptance in Nat. Commun. after some minor revisions.

Thanks for the encouraging comments.

1. Strictly speaking, β -ketoesters or β -ketoamides is not simple ketones. So please consider revising the title. Using β -ketocarboxyls instead of ketones might be OK, right?

Reply: We have changed “simple ketones” to “ β -ketocarboxyls” .

2. In Fig. 1, the word “1,3-axial central chirality” is overlapped with other chemical structures. Please revise it.

Reply: Thanks. This issue has been corrected.

3. Both the title and Fig.3C have indicated that aldehydes are part of the substrate scope, but I did not find any related description in the context. Am I missing something?

Reply: The missing paragraph has been supplemented in the text of the manuscript.

4. The π -allyl-palladium intermediate II is syn- η^3 rather than anti- η^3 . Please revise them accordingly throughout the manuscript.

Reply: Thanks. This has been modified.

5. Reference 22 is not completed.

Reply: Corrected

6. Given that synergistic catalysis enabled stereodivergent synthesis is developed rapidly, some updated literature could be considered to cite.

Reply: Additional references were cited in ref. 44-47.

REVIEWERS' COMMENTS

Reviewer #1 (Remarks to the Author):

The authors have completely revised related errors. Now I strongly support its publication.